# Mitochondrial NAD kinase Pos5 is required for CoQ biosynthesis in yeasts

Shogo Nishihara[1], Ikuhisa Nishida[2], Yasuhiro Matsuo[1,3,4], Tomohiro Kaino[1,3,4]\*, Makoto Kawamukai[3]\*

1 Bioresource and Life Sciences, The United Graduate School of Agricultural Sciences, Tottori University, Koyama-minami, Tottori, Japan, 2 Sakeology Center, Niigata University, Ikarashi, Niigata, Japan, 3 Department of Life Sciences, Faculty of Life and Environmental Sciences, Shimane University, Nishikawatsu, Matsue, Japan, 4 Institute of Agricultural and Life Sciences, Academic Assembly, Shimane University, Nishikawatsu, Matsue, Japan

\* kawamuka@life.shimane-u.ac.jp (MK); tkaino@life.shimane-u.ac.jp (TK)

## Abstract

Coenzyme Q (CoQ) is an essential component of the electron transport chain, and ten genes involved in CoQ biosynthesis have been identified in *Schizosaccharomyces pombe*. To gain further insight into CoQ biosynthesis, we screened the Bioneer gene-deletion library and found that the Δ*pos5* strain produced only 0.2-fold of the wild-type $CoQ_{10}$ level. Pos5 shares homology with *Saccharomyces cerevisiae* Pos5 (ScPos5), a mitochondrial NADH (or $NAD^+$) kinase that generates NADPH (or $NADP^+$). Heterologous expression of *ScPOS5* in the *S. pombe* Δ*pos5* strain recovered CoQ content to 0.9-fold of the wild-type level, indicating functional conservation of Pos5 between the two yeasts. Consistently, $CoQ_6$ level in Δ*Scpos5* was decreased to 0.2-fold of that in the wild-type strain. The Δ*pos5* strain exhibited several phenotypes characteristic of CoQ-deficient *S. pombe*, including inability to grow on non-fermentable carbon sources, hypersensitivity to oxidative stress, and high sulfide production. Among CoQ biosynthetic enzymes, Coq6 monooxygenase is thought to utilize NADPH. Supplementation with VA or PHB partially restored CoQ production in the Δ*pos5* strain, while overexpression of *coq6* had negligible effect. These findings suggest that Pos5 is required for the earlier step of CoQ biosynthesis.

## Introduction

### CoQ biosynthetic pathway

Coenzyme Q (CoQ), also known as ubiquinone, is an essential component of the respiratory chain required for energy production. CoQ cycles between reduced [$CoQH_2$] and oxidized [CoQ] forms [1]. This is a redox property important for electron transfer during respiration and for functioning as an antioxidant. Eukaryotes and bacteria belonging to the phylum Pseudomonadota synthesize CoQ endogenously, with species-specific variations in side chain length; for example, *Homo sapiens* and

**Data availability statement:** All relevant data are within the manuscript and its Supporting information files.

**Funding:** This work was partly supported by grant-in-aid funding from the Ministry of Education, Culture, Sports, Science, and Technology of Japan (#17H03806, #21H02117 and # 24K08715 to M. K.; #18K05393 to T. K.; #18K14377 to I. N.); the Mishima Kaiun Memorial Foundation to I. N.; the Science and Technology Research Promotion Program for Agriculture, Forestry, Fisheries, and Food Industry (#957613) to M. K.; we also thank the Faculty of Life and Environmental Sciences at Shimane University for the financial support for publishing this report. The funders had no role in study design, data collection and analysis, decision to publish, or preparation of the manuscript.

**Competing interests:** The authors have declared that no competing interests exist.

*Schizosaccharomyces pombe* produce $CoQ_{10}$, where the number of isoprene units is ten, *Saccharomyces cerevisiae* produces $CoQ_6$, and *Escherichia coli* produces $CoQ_8$ [2,3]. CoQ biosynthesis comprises mainly three stages: benzoquinone ring formation, isoprene side chain synthesis, and modification of the prenylated quinone ring [4]. The precursor of the side chain is synthesized from isopentenyl diphosphate and farnesyl diphosphate by polyprenyl diphosphate synthase [5]. Then, it is transferred to *p*-hydroxybenzoic acid (PHB) or *p*-aminobenzoic acid by *p*-hydroxybenzoate–polyprenyl diphosphate transferase (Coq2 or Ppt1) [6,7]. In eukaryotes, PHB is derived from tyrosine or other amino acids. The quinone ring of prenylated PHB then undergoes modifications, including methylations (Coq3 and Coq5), decarboxylation (Coq4), and hydroxylations (Coq6 and Coq7), to generate mature CoQ [2,8,9]. These reaction enzymes are encoded by nine genes in *S. cerevisiae* (*COQ1-COQ9*) and ten genes in *S. pombe* (*dps1*, *dlp1*, *ppt1*, and *coq3-coq9*) [10–14]. Those genes were utilized for $CoQ_{10}$ bioproduction in *S. pombe* [15]. In addition, benzoic acid inhibits the synthesis of CoQ [11], protein kinase A (Pka1) controls the level of CoQ [16], and regulatory factors such as Coq11 and Coq12 have recently been identified in *S. pombe*, suggesting that further more unknown factors are involved in regulating CoQ biosynthesis [17].

Among the deletion mutants that showed a lower $CoQ_{10}$ level in *S. pombe*, we selected the *pos5* mutants for further analysis. Although Pos5 has been extensively studied in *S. cerevisiae*, very little is known about its function in *S. pombe*. In *S. cerevisiae*, Pos5 is a unique mitochondrial nicotinamide adenine dinucleotide NAD(H) kinase that generates NADPH or $NADP^+$ from NADH or $NAD^+$ [18–22]. Because of its polarity, mitochondrial NADP(H) is synthesized from NAD(H) via mitochondrial NAD(H) kinase, as no mitochondrial transporter has been identified in yeast. In mitochondria, $NADP^+$ is essential for several processes, including the TCA cycle, amino acid biosynthesis, glutathione reduction, and Fe-S cluster biogenesis [19–22]. However, the relevance of NAD(H) kinase activity in CoQ biosynthesis has not been documented in any organism. Therefore, in this study, we focused on elucidating the role of Pos5 involves in CoQ synthesis.

## Materials and methods

### Yeast and *E. coli* strains, and growth media

Yeasts and *E. coli* strains used in this study are listed in Table 1. Yeast standard media and genetic manipulation methods have been described previously [23]. *S. pombe* strains were grown in complete YES medium (0.5% yeast extract (OXOID), 3% glucose, supplemented with 225 mg/mL adenine sulfate, 225 mg/mL leucine, 225 mg/mL uracil, 225 mg/mL histidine, and 225 mg/mL lysine hydrochloride). A non-fermentable carbon source medium, YEGES, containing 0.5% yeast extract, 2% glycerol, 1% ethanol, supplemented with 225 mg/mL adenine sulfate, 225 mg/mL leucine, 225 mg/mL uracil, 225 mg/mL histidine, and 225 mg/mL lysine hydrochloride, was used. PM medium comprised 0.3% potassium hydrogen phthalate, 0.56% sodium phosphate, 0.5% ammonium chloride, 2% glucose, and standard vitamins, minerals, and salts. PMGALU medium contained 0.375% glutamate as the nitrogen

**Table 1. Strain list.**

| Strain | Genotype | Resource |
|---|---|---|
| *S. pombe* | | |
| PR109 | *h⁻ leu1–32 ura4-D18* | Lab stock |
| PR110 | *h⁺ leu1–32 ura4-D18* | Lab stock |
| KH2 (OG1) | *h⁺ leu1–32 ura4-D18 ppt1::kanMX6* | Hayashi K. *et al*., 2014 |
| KH6 (PC976) | *h⁺ leu1–32 ura4-D18 coq6::kanMX6* | Hayashi K. *et al*., 2014 |
| LJ1030 | *h⁺ leu1–32 ura4-D18 dps1::kanMX6* | Zhang M. *et al*., 2008 |
| NSP7 | *h⁺ leu1–32:leu1-pJK148P41nmt1-ScPOS5 ura4-D18 pos5::kanMX6* | This study |
| NSP11 | *h⁺ leu1–32:leu1-pJK148P41nmt1-pos5 ura4-D18 pos5::kanMX6* | This study |
| NSP12 | *h⁺ leu1–32:leu1-pJK148P41nmt1-MTS36UTR1 ura4-D18 pos5::kanMX6* | This study |
| NSP13 | *h⁺ leu1–32:leu1-pJK148P41nmt1-UTR1 ura4-D18 pos5::kanMX6* | This study |
| NSP15 | *h⁺ leu1–32:leu1-pJK148P41nmt1-ΔMTS83pos5 ura4-D18 pos5::kanMX6* | This study |
| NSP16 | *h⁻ leu1–32 ura4-D18 pos5-GFP(S65T)-kanMX6* | This study |
| NSP23 | *h⁺ leu1–32:leu1-pJK148Pnmt1 ura4-D18* | This study |
| NSP25 | *h⁺ leu1-32:pJK148-Pnmt1-coq6 ura4-D18 pos5::kanMX6* | This study |
| NSP26 | *h⁺ leu1–32:leu1-pJK148Pnmt1 ura4-D18 pos5::kanMX6* | This study |
| NSP27 | *h⁺ leu1–32:leu1-pJK148Pnmt1 ura4-D18 coq6::kanMX6* | This study |
| NSP28 | *h⁺ leu1–32:leu1-pJK148Pnmt1-coq6 ura4-D18 coq6::kanMX6* | This study |
| NSP60 | *h⁺ leu1–32:leu1-pJK148Pnmt1-atd1 ura4-D18 pos5::kanMX6* | This study |
| RM3 | *h⁺ leu1–32 ura4-D18 cyc1::kanMX6* | Miki R. *et al*., 2008 |
| RYP7 | *h⁺ leu1–32 ura4-D18 pos5::kanMX6* | This study |
| | | |
| Bioneer disruptant (Ver. 4) | | |
| Δleu1 | *h⁺ ade6-M216 leu1–32 ura4-D18 leu1::kanMX4* | Kim D.U. *et al*., 2010 |
| Δpos5 | *h⁺ ade6-M216 leu1–32 ura4-D18 pos5::kanMX4* | Kim D.U. *et al*., 2010 |
| Δarg11 | *h⁺ ade6-M216 leu1–32 ura4-D18 arg11::kanMX4* | Kim D.U. *et al*., 2010 |
| | | |
| *S. cerevisiae* | | |
| BY4741 | *MATa his3Δ1 leu2Δ0 met15Δ0 ura3Δ0* | Lab stock |
| MK1601 | *MATα his3Δ1 leu2Δ0 lys2Δ0 ura3Δ0 pos5::kanMX4* | Kawai S. |
| Δcoq2 | *MATa his3Δ1 leu2Δ0 met15Δ0 ura3Δ0 coq2::kanMX4* | Multiple-System Atrophy Research Collaboration. 2013 |
| | | |
| *E. coli* | | |
| DH5α | *F⁻ Φ80dlacZΔM15 Δ(lacZYA-argF)U169 deoR recA1 endA1 hsdR17(rK⁻, mK⁺) phoA supE44 λ⁻ thi-1 gyrA96 relA1* | Lab stock |

source instead of ammonium chloride and was supplemented with adenine sulfate, leucine, and uracil in PM. *S. cerevisiae* strains were grown in YPD medium (1% yeast extract, 2% peptone, and 2% glucose). Synthetic defined (SD) medium (2% glucose and 6.7 g/L yeast nitrogen base without amino acids (BD Biosciences), containing 19 mg/L adenine sulfate; 76 mg/L each of arginine, histidine, lysine hydrochloride, methionine, uracil, and tryptophan; and 395 mg/L leucine). SD without glucose with glycerol (SD-C+glycerol) medium contained 3% glycerol and 6.7 g/L yeast nitrogen base without amino acids and the same amount of above amino acids, uracil, and adenine sulfate. SC medium consisted of 2% glucose and 6.7 g/L yeast nitrogen base without amino acids, supplemented with 19 mg/L adenine sulfate; 76 mg/L each of

alanine, arginine, asparagine, aspartate, cysteine, glutamine, glutamate, glycine, isoleucine, histidine, L-inositol, lysine, methionine, phenylalanine, proline, serine, threonine, tryptophan, tyrosine, uracil, valine; 7.6 mg/L *p*-aminobenzoic acid; and 395 mg/L leucine).

### Construction of *S. pombe* strains

The oligonucleotide primers used in this study are listed in S1 Table. *S. pombe pos5* on the chromosome was disrupted by replacing *pos5* with a selectable marker as previously described [24]. The 1.6-kb *kanMX6* module was amplified using flanking sequences corresponding to the 5' and 3' ends of *pos5*. Resistant colonies were selected on YES plates containing 100 mg/L G418, and *pos5* disruption was verified using colony PCR. DNA fragments of 500–600 bp corresponding to the 5' or 3' regions of the gene were amplified by PCR using pos5del-A and pos5del-B or pos5del-C and pos5del-D primer pairs (S1 Table). The amplicons were fused to the ends of the *kanMX6* module using PCR. The PR110 strain was transformed with the resulting *pos5::kanMX6* fragments to obtain the *pos5* disruptant. The chromosomal deletion of *pos5* was confirmed by PCR using the nb2 and pos5del-check primers. The obtained strain was designated as RYP7 (Δ*pos5*). Pos5-GFP-tagged strain was constructed using the recombinant PCR approach described in a previous study [24]. The pFA6a-GFP(S65T)-kanMX6 plasmid [24] was used as the template DNA, and the resulting PCR products carried the *GFP-kanMX6* cassette in the 3' region downstream of *pos5*. The oligonucleotides pos5-TAGW, pos5-TAGX, pos5-TAGY, and pos5-TAGZ were used to construct the *pos5-GFP-kanMX6* strain. The resulting *pos5-GFP-kanMX6* cassette was introduced into the PR109 strain, and the transformants carrying the *GFP*-fused *pos5* were verified by colony PCR [25]. The *S. cerevisiae* Δ*pos5* strain (Δ*Scpos5*; MK1601) was provided by Shigeyuki Kawai (Ishikawa Prefectural University).

### Plasmid construction

The plasmids used in this study were constructed by a method described previously (S2 Table) [10]. Each gene encoding NAD⁺/NADH kinase was PCR amplified using the *S. pombe* PR110 genome and the *S. cerevisiae* BY4741 genome as templates, with primers containing restriction sites. The amplified fragments were digested using restriction endonucleases and then inserted into the appropriate sites of the pREP41, pJK148-P$_{nmt1}$ or pJK148-P41$_{nmt1}$ vector by ligation. pREP41-pos5 was constructed by inserting the PCR product amplified using pos5(SalI)-F and pos5(BamHI)-R primers into the SalI and BamHI sites of pREP41. pREP41-coq6 was constructed by inserting the fragment digested from pREP1-coq6 by SalI and SmaI into the same sites of pREP41 [10]. Further, the other plasmids pREP41-ScPOS5 and UTR1 were also constructed similarly. To construct mitochondrial NAD kinase, mitochondrial-targeting sequence of *coq3* from *S. pombe* was fused to the *UTR1* sequence from *S. cerevisiae*. The mitochondrial transit peptide in *S. cerevisiae* Pos5p was 62 amino acids from the N-terminus, and its homologous position is 83 amino acids in *S. pombe* Pos5. Thus, the primers were designed to anneal at 298 bp from the 5'-terminus of Pos5. pJK148-Pnmt1 was constructed from pJK148 and pREP3X. The Pnmt1-MCS-Tnmt1 region was amplified and inserted into KpnI and SacI sites of pJK148. pJK148-P41nmt1 was constructed from pJK148 and pREP41X. The P41nmt1-MCS-Tnmt1 region was amplified and inserted into KpnI and SacI sites of pJK148. pJK148-P41nmt1-pos5 was constructed by inserting the *pos5* insert fragment digested from pREP41-pos5 into the SalI and BamHI sites of pJK148-P41nmt1. The other plasmids pJK148-P41nmt1-ScPOS5, UTR1, Spcoq3MTS36UTR1, ΔMTS83pos5, and pJK148-Pnmt1-coq6 were constructed similarly. To examine the cellular localization of Pos5, GFP fusion was generated by inserting *pos5* into the pSLF272L-GFP(S65A) vector [26,27]. pSLF272L-pos5-GFP(S65A) was constructed by inserting the PCR product amplified using the pos5-GFP(XhoI)-F and pos5-GFP(NotI)-R3 primers into XhoI and NotI sites of pSLF272L-GFP(S65A). The genes amplified by PCR were verified using DNA sequencing.

### CoQ extraction and measurement

Yeast precultures were inoculated into large-volume media and incubated for the indicated times. Unless otherwise specified, strains were grown at 30°C in 55 mL of liquid media (with or without specific supplements), starting from an

initial density of $1 \times 10^5$ cells/mL, and cultured for 48 or 72 hours. Cell numbers were counted using a Sysmex CDA-1000B (Sysmex, Tokyo, Japan), and the $OD_{600}$ was measured using a Shimadzu UVmini-1240 spectrophotometer (Shimadzu, Kyoto, Japan). Cells were harvested, and CoQ was extracted using the autoclave method as described previously [10]. Prior to extraction, 5 μg of $CoQ_6$ was added to each sample as an internal standard. Crude CoQ samples were separated by normal-phase thin-layer chromatography using a Kieselgel 60 $F_{254}$ plate (Merck Millipore, MA, USA). The TLC was developed with benzene as the solvent. After development, the TLC plate was visualized under UV illumination, and the bands corresponding to $CoQ_6$ and $CoQ_{10}$ were excised and extracted with hexane/isopropanol (1:1, v/v). The sample solvents were evaporated, and the dried solids were dissolved in ethanol. Purified CoQ samples were analyzed using high-performance liquid chromatography on a Shimadzu HPLC Class VP series instrument (Shimadzu). A reversed-phase YMC-Pack ODS-A column (A-312–3 AA12S03-1506PT, $150 \times 6$ mm, 3-μm particle size, 120 Å, YMC, Kyoto, Japan) was used. The mobile phase consisted of ethanol at a flow rate of 1.0 mL/min. $CoQ_6$, and $CoQ_{10}$ were detected by UV absorption at 275 nm.

## Isolation of mitochondria

Yeast cells were pre-cultured for 24 hours in 100 mL of YES medium and then inoculated into 3 L of YES medium. After incubation for 16–20 hours, cells were harvested at $OD_{600} = 1$. Mitochondria were isolated according to a previously described method [27] with slight modifications. In the current experiment, we incubated the pellet with 100 mM Tris-$SO_4$ and 10 mM DTT for 30 minutes at 30°C. To improve the yield of mitochondria, the pellet obtained by the initial homogenization and centrifugation was resuspended in a buffer containing 0.6 M mannitol, 20 mM HEPES-KOH, 0.5 mM EDTA, and 1 mM PMSF, and further homogenized 15 times.

## Measurement of NADP(H)

The concentrations of $NADP^+$ and NADPH in isolated yeast mitochondria extracts were determined using an enzymatic cycling assay according to a method reported previously [21,28,29]. Briefly, 50 μL of each sample was mixed with an equal volume of either 0.1 N HCl (for $NADP^+$ measurement) or 0.1 N KOH (for NADPH measurement), followed by incubation at 85°C for 3 min. Subsequently, the treated extracts and the corresponding $NADP^+$ or NADPH standards were added to a reaction mixture to a final volume of 200 μL containing 100 mM HEPES-KOH (pH 8.0), 0.5 mM EDTA, 2.5 mM glucose-6-phosphate (G6P), 1.66 mM phenazine ethosulfate, and 0.42 mM MTT (3-(4,5-dimethyl-2-thiazolyl)-2,5-diphenyl-2H-tetrazolium bromide). The reaction was initiated by the addition of 0.5 U of G6P dehydrogenase, and absorbance at 570 nm ($A_{570}$) was measured using a Corona SH-9000Lab microplate reader (Hitachi, Tokyo, Japan).

## Mitochondrial staining and fluorescence microscopy

Mitochondria were stained using the MitoTracker Red FM dye (Invitrogen, Thermo Fisher Scientific, Inc). Cells were suspended in PMU medium and incubated with 50 nM MitoTracker Red FM at room temperature for 1 hour. Imaging was performed at 1000x magnification using a BX2-FL-2 fluorescence microscope (Olympus). GFP(S65A) fluorescence was observed at an excitation wavelength of 485 nm. Fluorescent images were obtained using a DP74-SET-A digital camera (Olympus) connected to the microscope and processed using cellSens ver.2.2 (Olympus).

## Data and statistical analyses

Data from control and experimental samples were compared using the two-sample *t*-tests in Microsoft Excel (WA, USA). *p*-values <0.05 were considered statistically significant. Data from control and experimental samples were compared using one-way ANOVA with a post hoc test (Dunnett's test) performed with EZR (Jichi Medical University, Tochigi, Japan) [30]. EZR is a graphical user interface for R (The R Foundation for Statistical Computing, Vienna, Austria). More precisely, it is a modified version of R commander designed to add statistical functions frequently used in biostatistics.

                                                                                                 

# Results

## The *S. pombe* Δ*pos5* strain exhibits a phenotype similar to that of the CoQ-deficient strain

We have previously investigated the genes involved in CoQ biosynthesis of *S. pombe* using a Bioneer gene-deletion library and obtained approximately 40 individual gene-deleted strains with a $CoQ_{10}$ content lower than that of the wild-type strain [17]. In this study, we selected a Δ*pos5* strain from these strains for further analysis because it exhibited respiration deficiency, similar to CoQ-deficient strains, in addition to low $CoQ_{10}$ production. We independently constructed a Δ*pos5* strain to ensure that the phenotype observed in the Δ*pos5* strain from Bioneer Corp. is the same as our construct. CoQ levels in the Δ*pos5* strain of our construct were decreased to 0.2-fold of those in the wild-type strain (Fig 1A and 1B) as in the originally screened Bioneer Δ*pos5* strain (Fig 1C and 1D). Subsequently, we examined the phenotypes previously observed in CoQ-deficient strains of *S. pombe*, which exhibit respiratory deficiency, growth delay in minimal media, $H_2O_2$ sensitivity, and enhanced $H_2S$ production [13,31]. The Δ*pos5* strain failed to grow on YEGES medium containing glycerol and ethanol as non-fermentable carbon sources and showed retarded growth on YES containing hydrogen peroxide as well as on minimal medium (Fig 1E). Supplementation with arginine partially restored Δ*pos5* growth (S1 Fig) as observed previously [32]. This is due to the requirement of NADPH for Arg11-catalyzed reaction in arginine biosynthesis. When grown on YES containing $CuSO_4$, Δ*pos5* colonies also developed a brown coloration, similar to the Δ*dps1* strain, which is completely defective in $CoQ_{10}$ synthesis (Fig 1E). In addition, Δ*pos5* cells showed a round morphology, which is often seen in the mutants related to sexual differentiation [33]. The phenotype was reverted to the normal rod shape upon expression of *pos5* or mitochondrially targeted *UTR1*, which encodes a cytosolic NADK responsible for NADP(H) synthesis in *S. cerevisiae*, resembling the morphology of wild-type cells (S2 Fig). To determine whether the reduced CoQ level in the Δ*pos5* strain is simply a consequence of defective respiration, we next compared the CoQ content of the Δ*pos5* strain with that of a cytochrome *c*-deficient respiration mutant (Δ*cyc1*) (Fig 1F and 1G). The Δ*cyc1* strain did not show a marked decrease in CoQ levels, suggesting that respiratory deficiency alone does not account for the low CoQ level in the Δ*pos5* strain. These results indicate that Pos5 is specifically important in CoQ biosynthesis in *S. pombe*.

## Pos5 functions as an NAD(H) kinase

*S. pombe pos5* gene is predicted to encode a mitochondrial NAD(H) kinase because the Pos5 protein shares 37% identity with *S. cerevisiae* Pos5, a well-characterized mitochondrial NADH kinase (Fig 2A) [20,34]. To verify the functional similarity of SpPos5 and ScPos5, we constructed the Δ*pos5*+pJK148-P41nmt1-ScPOS5 strain (NSP7), in which ScPos5 was integrated at the chromosomal *leu1* locus of the *S. pombe* Δ*pos5* strain, and measured its CoQ content. CoQ levels in the NSP7 strain were recovered to 0.9-fold of that observed in the Δ*pos5*+*pos5* strain (NSP11) (Fig 2B and 2C), indicating functional similarity of these two proteins.

In the *S. cerevisiae* Δ*pos5* strain, mitochondrial NADP(H) levels are decreased [21]. To determine whether the *S. pombe* Δ*pos5* strain also influences mitochondrial NADP(H), we isolated mitochondria from the *S. pombe* Δ*pos5* strain and quantified NADP(H) content as described in materials and methods. Mitochondrial $NADP^+$ level and total NADP(H) level in Δ*pos5* were decreased to 0.6- and 0.8-fold, respectively, of those in the wild-type strain (Fig 3A and 3B), supporting that *S. pombe pos5* encodes NADP(H) kinase.

## ScPos5p is involved in CoQ biosynthesis in *S. cerevisiae*

The *S. cerevisiae* Δ*pos5* (Δ*Scpos5*) strain has previously been reported to exhibit respiratory deficiency, hydrogen peroxide sensitivity, and arginine auxotrophy [19]. We confirmed these phenotypes (Fig 4A). However, since the role of ScPos5 in CoQ biosynthesis has never been documented, we quantified CoQ levels in a Δ*Scpos5* strain. $CoQ_6$ levels in Δ*Scpos5* were decreased to 0.2-fold of those in the wild-type strain (Fig 4B and 4C), which is similar to the CoQ deficiency observed in the *S. pombe* Δ*pos5* mutant (Fig 1). Thus, ScPos5 is also involved in CoQ biosynthesis in *S. cerevisiae* to a similar extent as observed in the *S. pombe* Δ*pos5* strain.

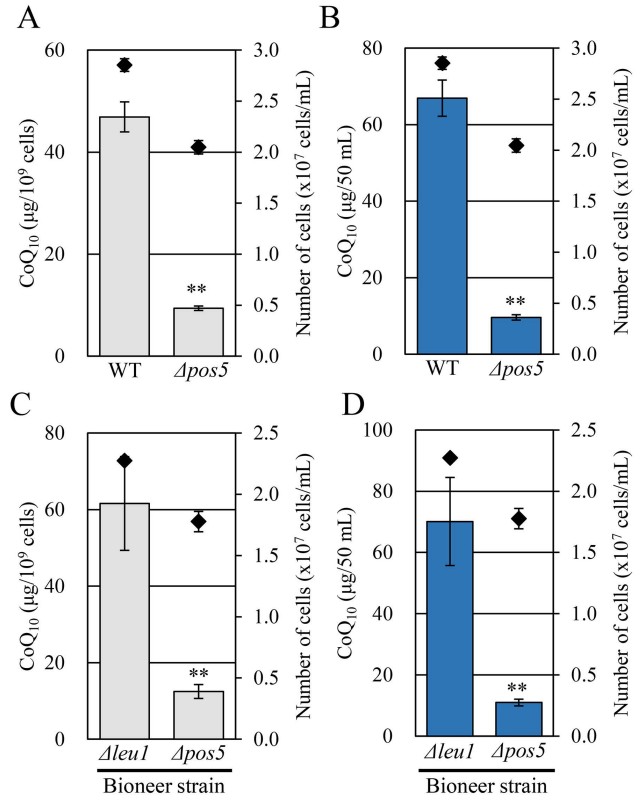

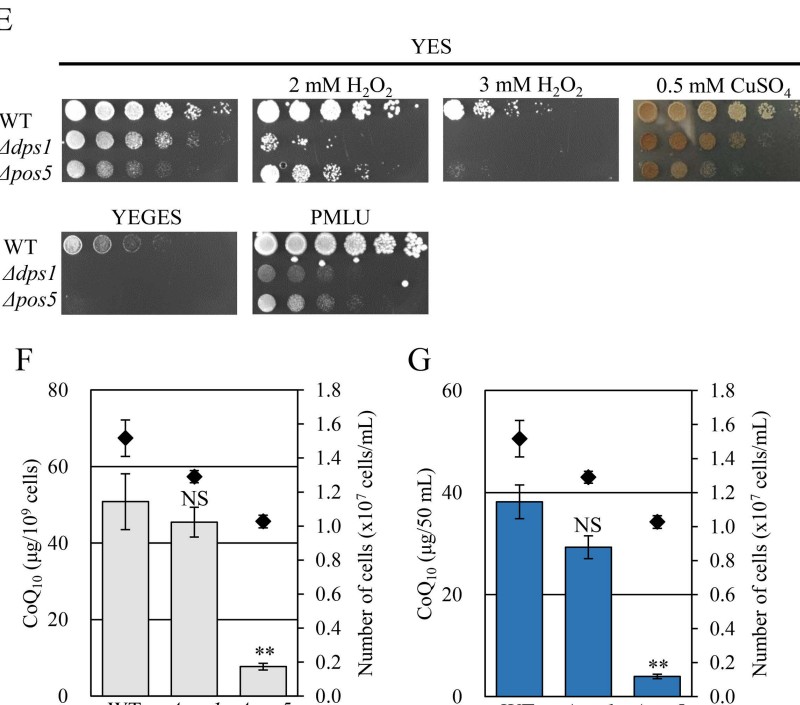

**Fig 1. The Δ*pos5* strain exhibits a phenotype similar to that of a CoQ-deficient strain.** A, B: CoQ$_{10}$ levels of wild-type and Δ*pos5* strains. Cells were cultured in YES medium for 48 hours. Diamonds (♦) show cell number. Bars indicate CoQ$_{10}$ content per cell (A) and per culture volume (B). Error bars indicate the S.D. of three independent measurements. \*\*: $p < 0.01$; statistical significance in CoQ levels (Student's *t*-test) versus wild-type strain. C,

D: CoQ$_{10}$ levels of the Δ*leu1* and Δ*pos5* strains obtained from Bioneer Corp. Cells were cultured in YES medium for 48 hours. Bars indicate the CoQ$_{10}$ content per cell (C) and per volume (D). Error bars indicate the S.D. of three independent measurements. **: $p < 0.01$; statistical significance in CoQ levels (Student's *t*-test) versus Δ*leu1* strain. E: Wild-type, Δ*dps1*, and Δ*pos5* strains were serially diluted (1:5) from 1 x 10$^7$ cells/mL and spotted onto YES, YEGES (2% glycerol and 1% ethanol), YES + 2, 3 mM H$_2$O$_2$, YES + 0.5 mM CuSO$_4$, and PMLU media. Plates were incubated at 30°C for 3–7 days (YES: 3 days, YEGES, YES + H$_2$O$_2$, YES + CuSO$_4$: 5 days, PMLU: 7 days). The Δ*dps1* strain, which is CoQ-deficient, was included for comparison. F & G: Comparison of CoQ levels between Δ*pos5* and Δ*cyc1* strains. Wild-type, Δ*cyc1*, and Δ*pos5* strains were cultured in YES medium for 48 hours. Bars indicate CoQ$_{10}$ content per cell (F) and per volume (G). Error bars indicate the S.D. of three independent measurements. **: $p < 0.01$; statistical significance in CoQ levels (Student's *t*-test) versus the wild-type strain. NS: no significant difference.

## Localization of NAD(H) kinase to mitochondria is required for CoQ biosynthesis

We next investigated the localization of the Pos5-GFP strain, in which Pos5-GFP was expressed from the *pos5* locus. The Pos5-GFP signal did not show the expected mitochondrial localization pattern (S3A Fig). In addition, the Pos5-GFP strain failed to maintain normal CoQ production and showed a CoQ level similar to the Δ*pos5* strain (S3B and S3C Fig), indicating a loss of Pos5 function by tagging GFP which probably interfered Pos5 function. Therefore, we constructed a pSLF272L-Pos5-GFP(S65A) plasmid to express Pos5-GFP exogenously and examined Pos5 localization by introducing it into the wild-type strain. The Pos5-GFP fluorescence overlapped with the MitoTracker Red FM signal (Fig 5), indicating that Pos5 localizes to mitochondria as a mitochondrial NAD(H) kinase. It also indicates Pos5-GFP retains partial functionality, because multicopy Pos5-GFP but not a single copy of that is functional.

Given the phenotypes of the *pos5* strain are specific for mitochondrial function, the mitochondrial localization of Pos5 NAD(H) kinase is thought to be essential for CoQ synthesis as shown in *S. cerevisiae* [20]. To directly examine the significance of mitochondrial localization of NAD(H) kinase in CoQ biosynthesis, we constructed pJK148-P41nmt1-ΔMTS83pos5, pJK148-P41nmt1-UTR1, and pJK148-P41nmt1-MTS36UTR1 plasmids. The Δ*MTS83pos5* construct is designed to express a Pos5 protein lacking the N-terminal 83 amino acids. The *UTR1* construct expresses a cytosolic NAD(H) kinase from *S. cerevisiae*. The MTS36UTR1 construct is designed to express a fusion protein comprising the N-terminal 36 amino acids of SpCoq3 fused to *S. cerevisiae* Utr1p. These constructs were introduced into the Δ*pos5* strains to generate Δ*pos5* + Δ*MTS83pos5* (NSP15), Δ*pos5* + *UTR1* (NSP13), and Δ*pos5* + *MTS36UTR1* (NSP12) strains. CoQ quantification showed that only *MTS36UTR1* restored CoQ production, whereas neither Δ*MTS83pos5* nor *UTR1* could recover CoQ levels in the Δ*pos5* strain (Fig 6A and 6B). Consistently, a mitochondrial-targeted Utr1 restored CoQ levels in the Δ*pos5* strain (S4 Fig) and Utr1-GFP fusion (mito-UTR1-GFP) localized correctly to mitochondria and restored CoQ levels in such a strain (S5 Fig). Thus, these results demonstrate that cytosolic NAD(H) kinase from *S. cerevisiae* can replace the function of mitochondrial NAD(H) kinase when it is expressed in mitochondria, indicating that the localization of NAD(H) kinase to mitochondria is critical for CoQ biosynthesis.

## Vanillic acid and PHB partially restored CoQ content in the Δ*pos5* strain

Among the reactions in CoQ biosynthesis, Coq6 catalyzes C5-hydroxylation of the quinone precursor and requires reducing equivalents from NAD(P)H, through ferredoxin and ferredoxin reductase [35–37]. Ferredoxin is reduced by ferredoxin reductase utilizing NAD(P)H to provide electrons to Coq6 reaction in *S. cerevisiae*. In *S. pombe*, the ferredoxin reductase Arh1 utilizes both NADPH and NADH [38]. Based on these observations, we hypothesized Coq6 activity may be impaired in the Δ*pos5* strain. To test this, we overexpressed *coq6* in the Δ*pos5* strain and measured CoQ content. However, CoQ levels in the Δ*pos5* strain overexpressing *coq6* were comparable to that in the Δ*pos5* strain integrating a vector (Fig 7A and 7B).

In contrast, the Δ*coq6* strain expressing *coq6* on the chromosome clearly restored the CoQ level (S6 Fig). Because exogenous vanillic acid (VA) is known to restore CoQ levels in the Δ*coq6* strain (Fig 8A) [17], we added VA to the Δ*pos5* strain. VA clearly increased CoQ levels in the Δ*coq6* strain (Fig 8B and 8C). Although statistical difference was not

   

A

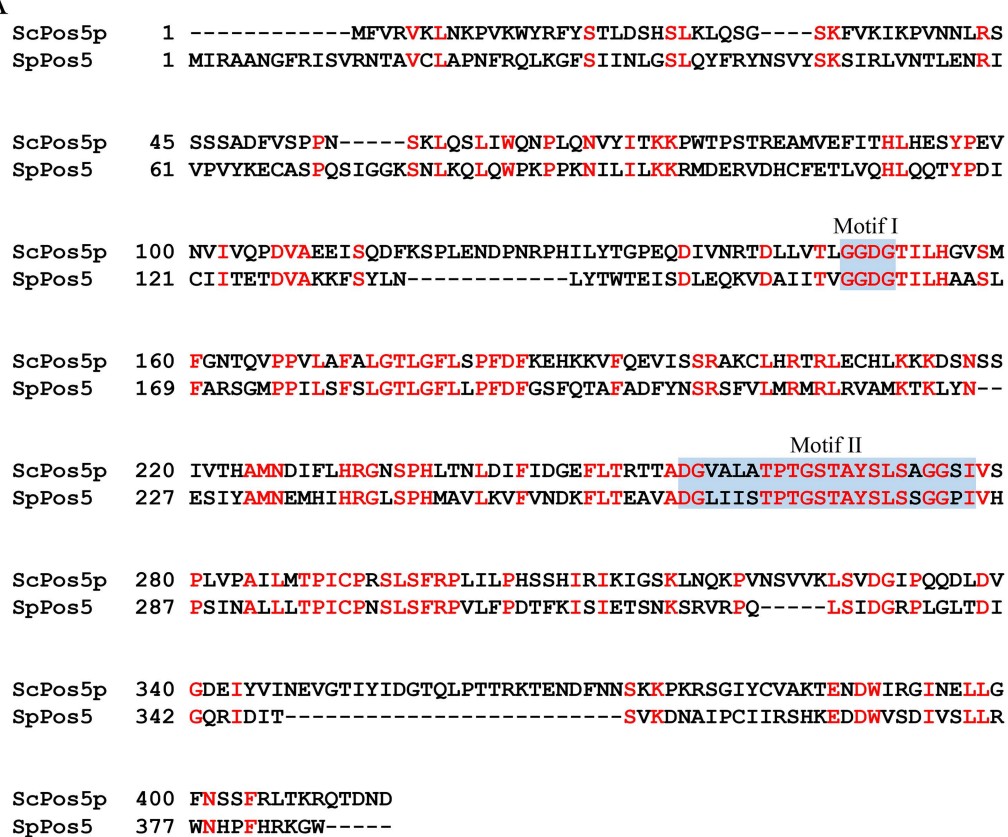

```
ScPos5p    1  -----------MFVRVKLNKPVKWYRFYSTLDSHSLKLQSG----SKFVKIKPVNNLRS
SpPos5     1  MIRAANGFRISVRNTAVCLAPNFRQLKGFSIINLGSLQYFRYNSVYSKSIRLVNTLENRI

ScPos5p   45  SSSADFVSPPN-----SKLQSLIWQNPLQNVYITKKPWTPSTREAMVEFITHLHESYPEV
SpPos5    61  VPVYKECASPQSIGGKSNLKQLQWPKPPKNILILKKRMDERVDHCFETLVQHLQQTYPDI

                                                          Motif I
ScPos5p  100  NVIVQPDVAEEISQDFKSPLENDPNRPHILYTGPEQDIVNRTDLLVTLGGDGTILHGVSM
SpPos5   121  CIITETDVAKKFSYLN-----------LYTWTEISDLEQKVDAIITVGGDGTILHAASL

ScPos5p  160  FGNTQVPPVLAFALGTLGFLSPFDFKEHKKVFQEVISSRAKCLHRTRLECHLKKKDSNSS
SpPos5   169  FARSGMPPILSFSLGTLGFLLPFDFGSFQTAFADFYNSRSFVLMRMRLRVAMKTKLYN--

                                                          Motif II
ScPos5p  220  IVTHAMNDIFLHRGNSPHLTNLDIFIDGEFLTRTTADGVALATPTGSTAYSLSAGGSIVS
SpPos5   227  ESIYAMNEMHIHRGLSPHMAVLKVFVNDKFLTEAVADGLIISTPTGSTAYSLSSGGPIVH

ScPos5p  280  PLVPAILMTPICPRSLSFRPLILPHSSHIRIKIGSKLNQKPVNSVVKLSVDGIPQQDLDV
SpPos5   287  PSINALLLTPICPNSLSFRPVLFPDTFKISIETSNKSRVRPQ-----LSIDGRPLGLTDI

ScPos5p  340  GDEIYVINEVGTIYIDGTQLPTTRKTENDFNNSKKPKRSGIYCVAKTENDWIRGINELLG
SpPos5   342  GQRIDIT----------------------SVKDNAIPCIIRSHKEDDWVSDIVSLLR

ScPos5p  400  FNSSFRLTKRQTDND
SpPos5   377  WNHPFHRKGW-----
```

B  C

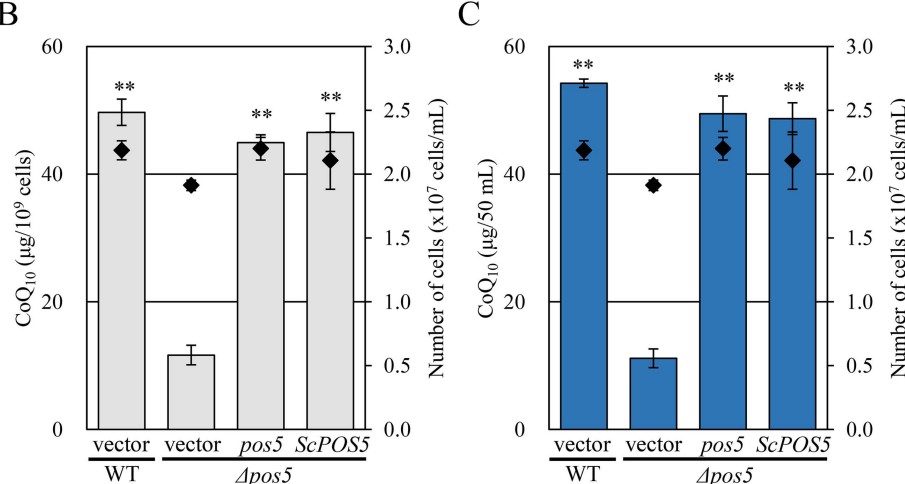

**Fig 2. *ScPOS5* overexpression restores CoQ levels in the *S. pombe* Δ*pos5* strain.** A: Sequence alignment of the Pos5 amino acid sequences from *S. pombe* (L972) and *S. cerevisiae* (S288C). Alignment was performed using ClustalW and visualized with the boxshade server. Conserved NAD kinase regions (I and II) are indicated by blue box. Motif I (GGDG) is part of the ATP-binding site, and Motif II represents a nucleotide-binding site. B, C: Restoration of CoQ$_{10}$ level in the Δ*pos5* strain by *ScPOS5* overexpression. Wild-type+vector (NSP23), Δ*pos5*+vector (NSP26), Δ*pos5*+*pos5* (NSP11), and Δ*pos5*+*ScPOS5* (NSP7) strains were cultured in YES medium for 48 hours. Diamonds (♦) show cell number. Bars indicate CoQ$_{10}$ content per cell (B) and per volume (C). Error bars indicate the S.D. of three independent measurements. **: $p < 0.01$; statistical significance in CoQ levels (Dunnett's test) versus Δ*pos5*+vector strain.

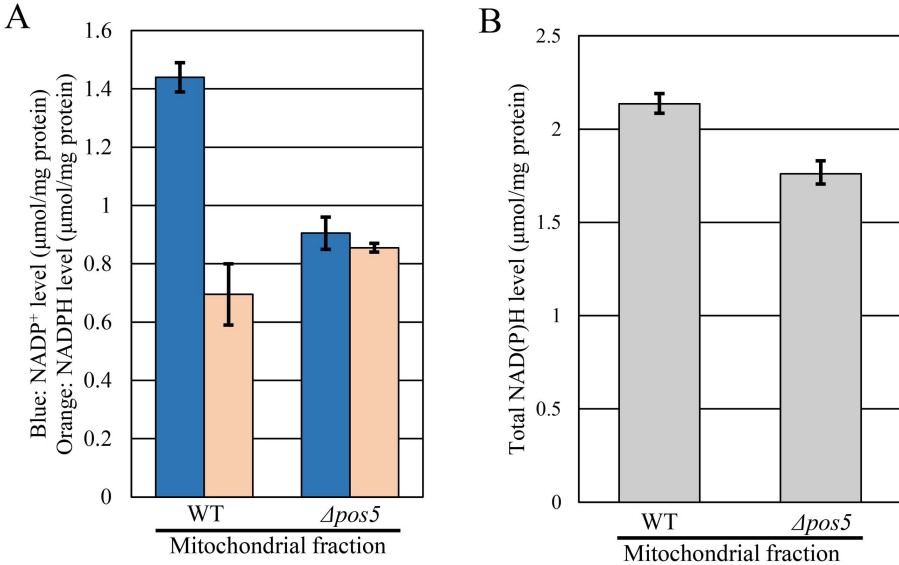

**Fig 3. The Δ*pos5* strain exhibits decreased NADP(H) levels.** A, B: Wild-type and Δ*pos5* strains were cultured in 3 L YES liquid medium and harvested at mid-log phase. Cell pellets were treated with DTT and Zymolyase for cell wall degradation. Spheroplasts were homogenized and centrifuged to obtain mitochondrial fractions. Protein concentrations in mitochondria-enriched fractions were quantified using the Bradford method. Mitochondrial NADP(H) concentrations were measured enzymatically using glucose-6-phosphate dehydrogenase. A: Quantification of mitochondrial $NADP^+$ and NADPH in wild-type and Δ*pos5* strains. B: Total NADP(H) levels presented.

observed, the addition of VA tended to increase CoQ levels in the Δ*pos5* strain (Fig 8D and 8E). These results suggest that Coq6 is not fully functional in the Δ*pos5* strain, but that impaired Coq6 activity is not the sole reason for the decreased CoQ content.

We next examined the effect of *p*-hydroxybenzoate (PHB), a quinone precursor, on CoQ production in the Δ*pos5* strain. PHB is condensed with decaprenyl diphosphate by Ppt1 to synthesize decaprenyl-PHB, which subsequently undergoes modifications to generate $CoQ_{10}$ [39]. Supplementation of 0.5 mM PHB partially increased CoQ levels in the Δ*pos5* strain compared to the untreated condition (Fig 8F and 8G), suggesting that NADP(H) availability affects a reaction upstream of CoQ biosynthesis.

We then tested overexpression of the *atd1* gene, which encodes a potential enzyme that converts *p*-hydroxybenzaldehyde to PHB, in the Δ*pos5* strain to see any effect on CoQ biosynthesis. The result showed slight increased CoQ levels in such a strain comparing with the one without the *atd1* expression (S7 Fig), but the difference was not statistically significant.

## Discussion

In this study, we showed that the mitochondrial NAD(H) kinase Pos5 is required for proper CoQ biosynthesis in both *S. pombe* and *S. cerevisiae*. In the Δ*pos5* strains of both species, CoQ levels were reduced to approximately 20% of the wild-type levels, indicated that Pos5 is important but not essential for CoQ biosynthesis. This observation indicates that the role of Pos5 in CoQ biosynthesis is different from the indispensable CoQ biosynthesis genes such as *dps1*, *dlp1*, and *coq2* to *coq9*, which are involved in the synthesis of prenyl tail and modification of the quinone ring precursor in *S. pombe*. Previous genetic screening in the *S. pombe* mutant have identified *coq11* and *coq12* as nonessential but functionally important for CoQ production [17]. Thus, *pos5*, *coq11*, and *coq12* are categorized as the factors that significantly affect CoQ levels without being absolutely required for CoQ synthesis. Because CoQ is indispensable for human survival,

A

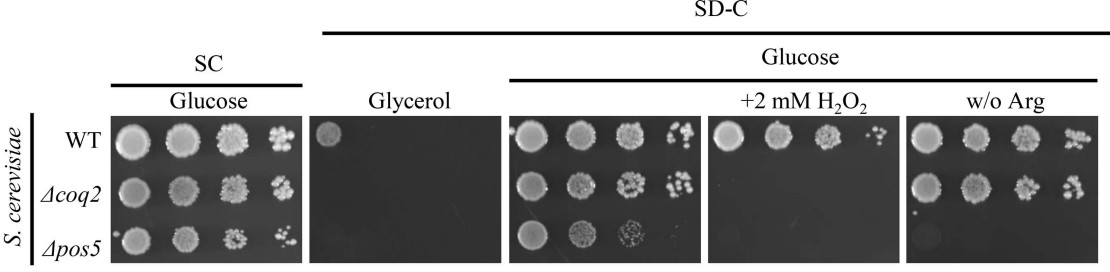

B C

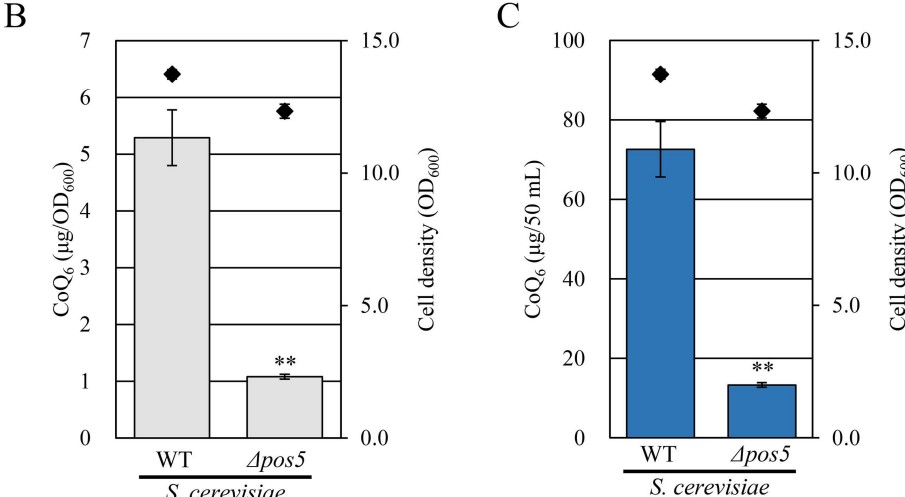

**Fig 4. The *S. cerevisiae* Δ*pos5* strain exhibits a CoQ-deficient phenotype.** A: *S. cerevisiae* wild-type, Δ*coq2*, and Δ*pos5* strains were serially diluted (1:10) from an initial $OD_{600}=2$, spotted onto the indicated media, and incubated at 30°C for several days (SC, SD (glucose), SD (glucose)+$H_2O_2$, SD (glucose) without arginine: 3 days. SD (glycerol): 6 days). The Δ*coq2* strain was included as a representative CoQ-deficient strain. B, C: $CoQ_6$ quantification in the *S. cerevisiae* Δ*pos5* strain. Wild-type and Δ*pos5* strains were cultured in YPD medium for 48 hours, starting from an initial $OD_{600}=0.02$. Diamonds (♦) show cell number. Bars indicate $CoQ_6$ content per cell (B) and per volume (C). Error bars indicate the S.D. of three independent measurements. **: $p < 0.01$; statistical significance in CoQ levels (Student's *t*-test) versus wild-type strain.

individuals who harbor mutations reducing CoQ production to ~20% of normal levels suffer severe damage in muscle, brain, and kidney tissues [9]. Therefore, identifying genes that are involved in CoQ biosynthesis is critical for understanding human genetic disorders associated with CoQ levels. Given that humans possess a mitochondrial NAD(H) kinase [40], exploring its relevance in CoQ biosynthesis is important for future research.

Pos5 is a mitochondrial NAD(H) kinase. This has been shown in *S. cerevisiae* Pos5 by *in vitro* assays demonstrating that purified Pos5 phosphorylates $NAD^+$ and NADH, with considerably higher NADH kinase activity [20,34]. Introduction of the *S. cerevisiae POS5* gene in the *S. pombe pos5* mutant restored CoQ production, supporting the idea that Pos5 is also an NAD(H) kinase. In *S. cerevisiae*, wild-type mitochondria contain approximately four times as much NADPH as the *pos5* mutant mitochondria and 2.5 times as much $NADP^+$ [21]. In contrast, in the *S. pombe* Δ*pos5* strain, we observed a reduction in total NADP(H) levels, with $NADP^+$ showing the most pronounced reduction. This may be due to the species difference.

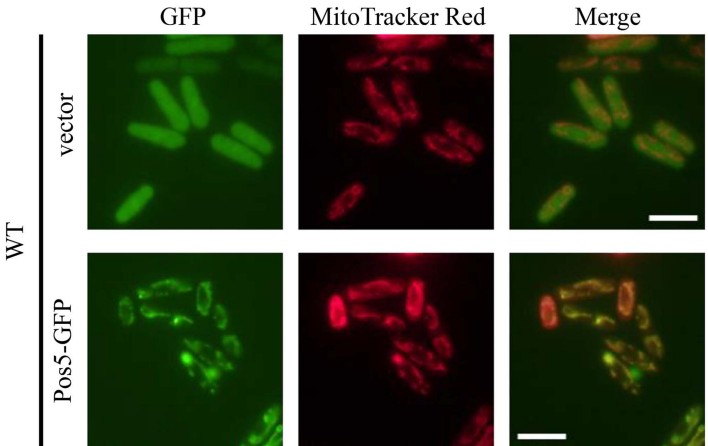

**Fig 5. Localization analysis of Pos5-GFP.** Wild-type harboring pSLF272L-GFP or pSLF272L-pos5-GFP cells were incubated in PMU medium containing 0.1 μM thiamine for 8 hours. Cells were collected at mid-log phase, stained with MitoTracker Red for 1 hour, washed, and observed by fluorescent microscopy. White bars indicate a scale of 10 μm.

The *pos5* deletion mutant exhibited several phenotypes, including respiratory deficiency, sensitivity to hydrogen peroxide, growth delay on minimal media, requirement of arginine for growth, elevated $H_2S$ production, a rounded cell morphology and reduced CoQ levels. The *S. cerevisiae* Δ*pos5* strain showed similar phenotypes except for $H_2S$ production [20,22]. In *S. pombe*, excessive $H_2S$ is produced caused by non-functionality of the sulfide quinone reductase (Hmt2) [41], which oxidizes sulfide using CoQ. Because *S. cerevisiae* lacks a similar enzyme, sulfide accumulation is not enhanced by CoQ deficiency. Mitochondrial NADP(H) produced by the NAD(H) kinase Pos5 is essential for maintaining the electron transfer system, presumed by the instability of Fe-S cluster proteins within the complex II and III components [22]. When we tested CoQ levels in a respiration-deficient mutant (the Δ*cyc1* strain), the CoQ levels were not drastically decreased. While we cannot rule out a possibility that Pos5 deficiency indirectly affects CoQ levels via impaired Fe-S cluster biogenesis, deficiency of respiration itself is not a cause of lower CoQ levels in the Δ*pos5* strain. Our observation that PHB restores CoQ biosynthesis implies that the core biosynthetic machinery downstream of PHB is functional.

We showed that the expression of a cytosolic NAD kinase in mitochondria restored CoQ levels in *S. pombe* Δ*pos5*, indicating that a sufficient mitochondrial NADPH pool is necessary for CoQ biosynthesis. Because Coq6 uses reducing equivalents of NADPH via ferredoxin and ferredoxin reductase [37,42], we initially hypothesized that Coq6 activity is limiting in the Δ*pos5* strain. However, overexpression of *coq6* in the Δ*pos5* strain did not restore the CoQ level. By contrast, the addition of the quinone precursor analog VA and PHB partially increased the CoQ level in Δ*pos5*. This suggests that the primary defect in the Δ*pos5* strain lies in the synthesis of the quinone precursor. In the quinone precursor synthesis pathway in *S. pombe*, the aldehyde dehydrogenase Atd1 is thought to catalyze the conversion of *p*-hydroxybenzaldehyde to PHB with NADH or NADPH reduction. Although overexpression of *atd1* in the Δ*pos5* strain did not clearly increase the CoQ levels, we observed a slight positive effect. Based on our findings, we propose that NADPH availability affects quinone precursor synthesis.

In conclusion, we found that the mitochondrial NAD(H) kinase Pos5 is critical in CoQ biosynthesis in both budding and fission yeasts. Our results suggest that the requirement for NADPH lies in the synthesis of the precursor of CoQ biosynthesis, although more detailed analysis is necessary to define the specific reaction(s) that depend on mitochondrial NADPH.

A

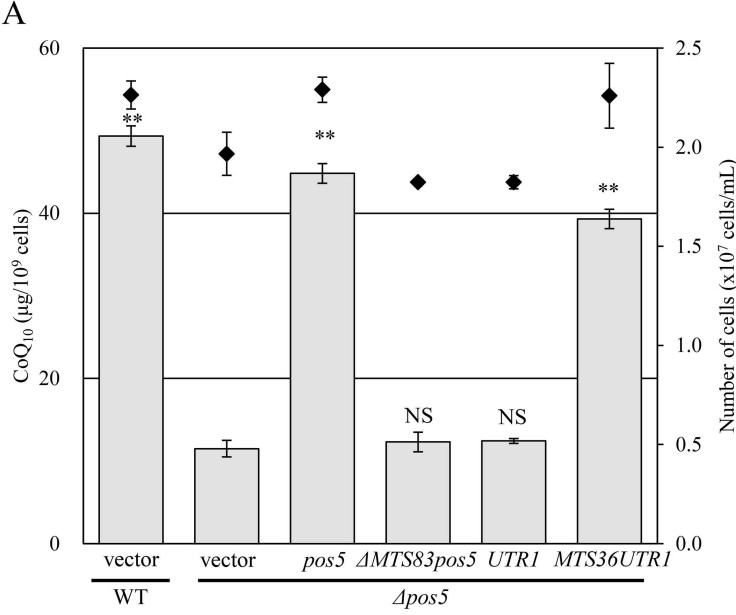

B

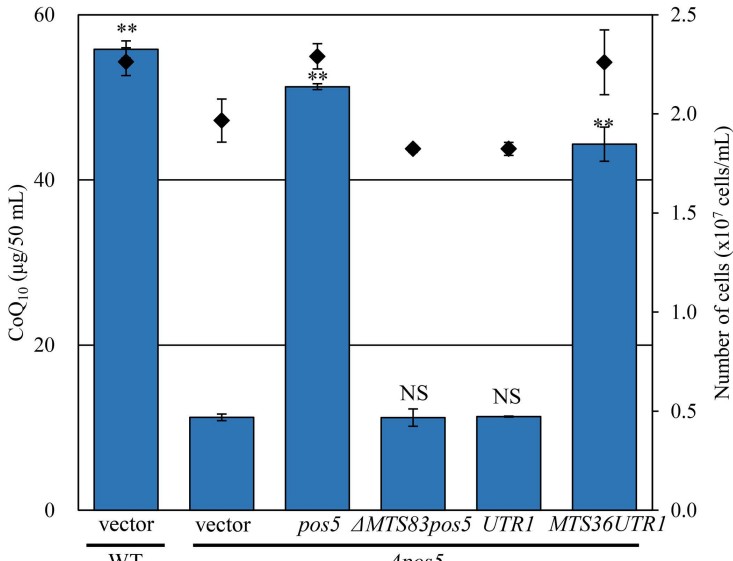

**Fig 6. Budding yeast NAD kinase targeted to mitochondria restores CoQ$_{10}$ levels in the Δ$pos5$ strain.** A, B: Wild-type+vector (NSP23), Δ$pos5$+vector (NSP26), Δ$pos5$+$pos5$ (NSP11), Δ$pos5$+Δ$MTS83pos5$ (NSP15), Δ$pos5$+$UTR1$ (NSP13), and Δ$pos5$+$MTS36UTR1$ (NSP12) strains were cultured in YES medium for 48 hours. Diamonds (♦) show cell number. Bars indicate CoQ$_{10}$ content per cell (A) and per volume (B). Error bars indicate the S.D. of three independent measurements. **: $p < 0.01$; statistical significance in CoQ levels (Dunnett's test) versus the Δ$pos5$+vector strain. NS: no significant difference versus the Δ$pos5$+vector strain.

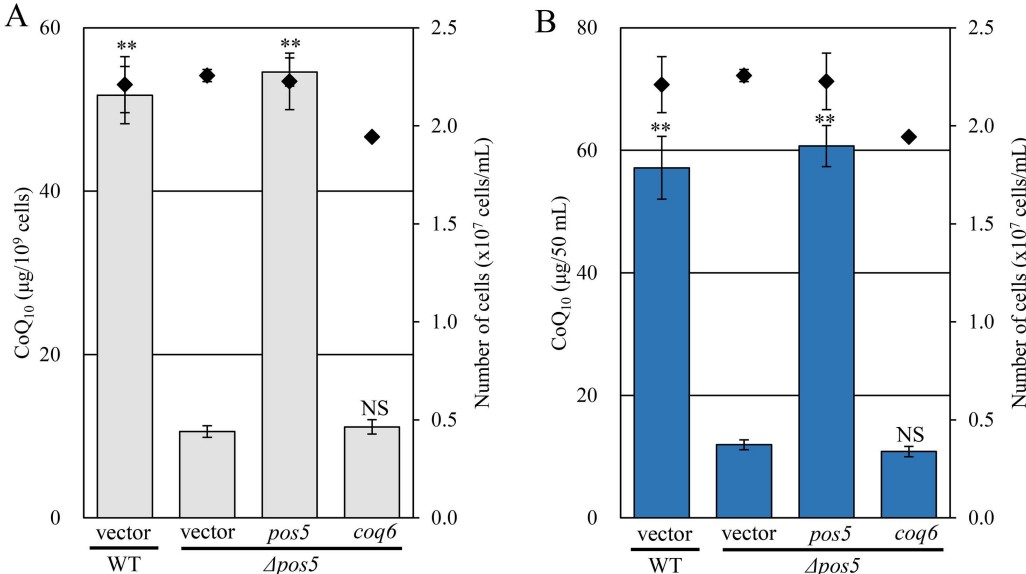

**Fig 7. Overexpression of *coq6* does not increase CoQ$_{10}$ levels in the Δ*pos5* strain.** Wild-type strain integrating the vector (NSP23), Δ*pos5* integrating the vector (NSP26), Δ*pos5* expressing *pos5* (NSP11), and Δ*pos5* expressing *coq6* (NSP25) were cultured in YES medium for 48 hours. Diamonds (♦) show cell number. Bars indicate CoQ$_{10}$ content per cell (A) and per volume (B). Error bars indicate the S.D. of three independent measurements. **: $p < 0.01$; statistical significance in CoQ levels (Dunnett's test) versus the Δ*pos5*+vector strain. NS: no significant difference with the Δ*pos5*+vector strain.

## Supporting information

**S1 Fig. The *S. pombe* Δ*pos5* strain exhibits arginine auxotrophy.** Wild-type, Δ*arg11*, and Δ*pos5* strains were serially diluted (1:5) from 1 x 10$^7$ cells/mL and spotted onto YES, PMGALU, and PMGALU+0.4 mg/mL arginine media. Plates were incubated at 30°C for 4 days. The Δ*arg11* strain, an arginine auxotroph, was included for comparison.
(TIFF)

**S2 Fig. Morphological analysis of the Δ*pos5* strain expressing NADK.** Wild-type+vector (NSP23), Δ*pos5*+vector (NSP26), Δ*pos5*+*pos5* (NSP11), Δ*pos5*+Δ*MTS83pos5* (NSP15), Δ*pos5*+*UTR1* (NSP13), and Δ*pos5*+*MTS36UTR1* (NSP12) strains were grown at 30°C in PMLU to the mid-logarithmic phase. The cells were resuspended in PMLU and observed using a BX2-FL-2 microscope (Olympus). The scale bars indicate 10 μm.
(TIFF)

**S3 Fig. Genomic Pos5 tagged with GFP does not localize properly to mitochondria and shows reduced CoQ levels.** A: Localization analysis of the Pos5-GFP strain. Pos5-GFP cells were collected at mid-log phase and stained with MitoTracker Red for 1 hour. After washing, the cells were examined using fluorescence microscopy. B, C: CoQ$_{10}$ quantification of the Pos5-GFP strain. Wild-type, Δ*pos5*, and Pos5-GFP strains were cultured in YES for 48 hours. Diamonds (♦) show cell number. Bars indicate CoQ$_{10}$ content per cell (B) and per volume (C). Error bars indicate the S.D. of three measurements. **: $p < 0.01$; statistical significance in CoQ levels (Dunnett's test) versus the Δ*pos5* strain.
(TIFF)

**S4 Fig. Restoration of CoQ in the *S. pombe* Δ*pos5* strain by expression of mitochondrially localized *S. cerevisiae* Utr1 on the plasmid.** A, B: Wild-type and Δ*pos5* strains harboring pREP41, pREP41-pos5, pREP41-UTR1, or

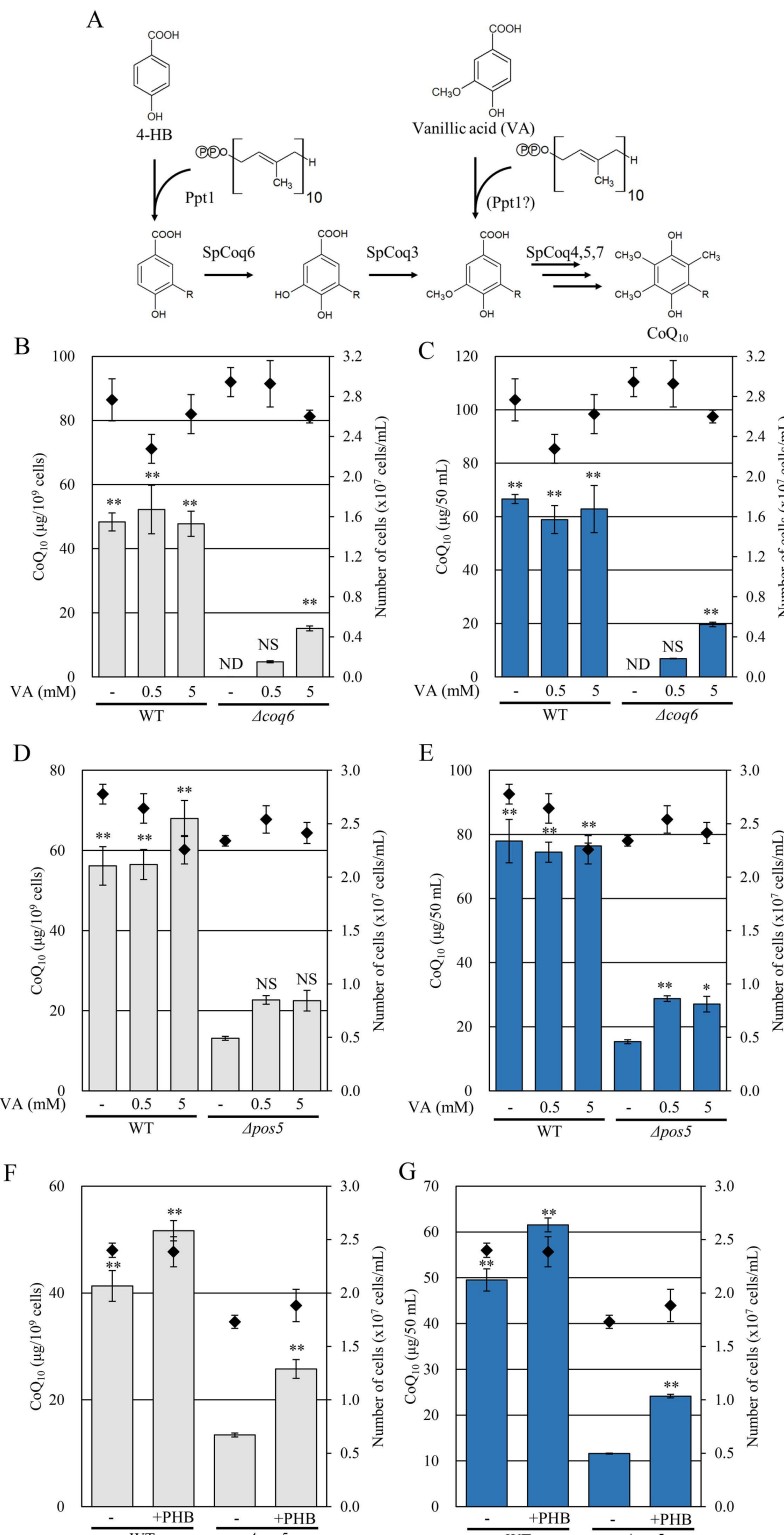

**Fig 8. Addition of VA and PHB increase CoQ₁₀ levels in the Δ*pos5* strain.** A: Schematic of the CoQ biosynthetic pathway in *S. pombe* and the quinone precursors used. PHB is the substrate for the early steps of CoQ biosynthesis. VA bypasses the reactions catalyzed by Coq6 and Coq3. -R indicates the decaprenyl moiety. B, C, D & E: Effect of VA to CoQ levels in Δ*coq6* and Δ*pos5* strains. B, C: Wild-type and Δ*coq6* strains were cultured in

YES and YES+VA (0.5 or 5 mM) medium for 48 hours. Diamonds (♦) show cell number. Bars indicate CoQ$_{10}$ content per cell (B) and per volume (C). Error bars indicate the S.D. of three measurements. **: $p < 0.01$; statistical significance in CoQ levels (Dunnett's test) versus the Δcoq6 strain. NS: no significant difference with the Δcoq6 strain. ND: not detected. D, E: Wild-type and Δpos5 strains were cultured in YES and YES+VA (0.5 or 5 mM) for 48 hours. Bars indicate CoQ$_{10}$ content per cell (D) and per volume (E). Error bars indicate the S.D. of three measurements. **: $p < 0.01$; *: $p < 0.05$ statistical significance in CoQ levels (Dunnett's test) versus the Δpos5 strain. NS: no significant difference with the Δpos5 strain. F & G: Effect of PHB to CoQ levels in the Δpos5 strain. Wild-type and Δpos5 strains were cultured in YES and YES+PHB (0.5 mM) medium for 48 hours. Diamonds (♦) show cell number. Bars indicate CoQ$_{10}$ content per cell (F) and per volume (G). Error bars indicate the S.D. of three measurements. **: $p < 0.01$; statistical significance in CoQ levels (Dunnett's test) versus the Δpos5 strain.

pREP41-MTS36UTR1 were cultured in PMU medium for 72 hours. Diamonds (♦) show cell number. Bars indicate CoQ$_{10}$ content per cell (A) and per volume (B). Error bars indicate the S.D. of two measurements. (TIFF)

**S5 Fig. Restoration of CoQ in the *S. pombe* Δpos5 strain by expression of mitochondrially localized *S. cerevisiae* Utr1 tagged with GFP.** A: Fluorescent microscopy of the Δpos5 strain expressing mitochondrial or cytosolic NAD$^+$/NADH kinase tagging with GFP at the C-terminus. Wild-type and Δpos5 strains harboring pSLF272L-GFP(S65A) or pSLF272L-UTR1-GFP(S65A) were grown at 30°C in 10 mL PMU, while Δpos5 strains harboring pSLF272L-pos5-GFP(S65A) and pSLF272L-MTS36UTR1-GFP(S65A) were grown at 30°C in 10 mL PMU + 0.1 μM thiamine. Cells were collected at 8 hours after inoculation from 5 x 10$^5$ cells/mL and stained with MitoTracker Red. The scale bar indicates 10 μm. B & C: Wild-type and Δpos5 strains harboring pSLF272L-GFP(S65A), pSLF272L-pos5-GFP(S65A), pSLF272L-UTR1-GFP(S65A), or pSLF272L-MTS36UTR1-GFP(S65A) were cultured in PMU medium for 72 hours. Diamonds (♦) show cell number. Bars indicate CoQ$_{10}$ content per cell (B) and per volume (C). Error bars indicate the S.D. of three measurements. *: $p < 0.05$; statistical significance in CoQ levels (Dunnett's test) versus the Δpos5 strain expressing GFP. NS: no significant difference. ND: not detected. D: Plasmid map of pSLF272L-pos5-GFP(S65A), pSLF272L-UTR1-GFP(S65A), and pSLF272L-MTS36UTR1-GFP(S65A). The vector pSLF272L contains *Pnmt41*, GFP(S65A), and *Tnmt1*. (TIFF)

**S6 Fig. Genomic integration of pJK148-Pnmt1-coq6 restores CoQ production in the Δcoq6 strain.** A, B: Wild-type strains integrated with the vector, Δcoq6 strains integrated with the vector, and Δcoq6 strains integrated with pJK148-Pnmt1-coq6 were cultured in YES medium for 48 hours. Diamonds (♦) show cell number. Bars indicate CoQ$_{10}$ content per cell (A) and per volume (B). Error bars indicate the S.D. of three measurements. **: $p < 0.01$; statistical significance in CoQ levels (Dunnett's test) versus the Δcoq6 strain integrated with the vector. ND: not detected. (TIFF)

**S7 Fig. Over-expression of *atd1* in Δpos5.** A, B: Wild-type+vector (NSP23), Δpos5+vector (NSP26), Δpos5+pos5 (NSP11), and Δpos5+atd1 (NSP60) strains were cultured in YES medium for 48 hours. Diamonds (♦) show cell number. Bars indicate CoQ$_{10}$ content per cell (A) and per volume (B). Error bars indicate the S.D. of three measurements. **: $p < 0.01$; statistical significance in CoQ levels (Dunnett's test) versus Δpos5 strain integrated with vector. NS: no significant difference. (TIFF)

**S1 Table. Primer list.**
(XLSX)

**S2 Table. Plasmid list.**
(XLSX)

## Acknowledgments

We thank Dr. T. Ogawa (Shimane University) for his help in NADPH and NADP⁺ measurements, Dr. S. Kawai (Ishikawa Prefectural University) for providing an *S. cerevisiae pos5* strain, and Dr. Y. Tamura (Yamagata University) for his advice on the isolation of mitochondria from *S. pombe*. We thank R. Yanai for constructing the *pos5* deletion strain and H. Sumi for her assistance.

## Author contributions

**Conceptualization:** Shogo Nishihara, Ikuhisa Nishida, Yasuhiro Matsuo, Tomohiro Kaino, Makoto Kawamukai.

**Formal analysis:** Shogo Nishihara.

**Funding acquisition:** Ikuhisa Nishida, Tomohiro Kaino, Makoto Kawamukai.

**Investigation:** Shogo Nishihara, Ikuhisa Nishida.

**Methodology:** Ikuhisa Nishida.

**Project administration:** Tomohiro Kaino.

**Supervision:** Yasuhiro Matsuo, Tomohiro Kaino, Makoto Kawamukai.

**Writing – original draft:** Shogo Nishihara, Makoto Kawamukai.

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
