## [Decision Letter · Decision Letter 0]

9 Dec 2025

Dear Dr. Kawamukai,

Thank you for submitting your manuscript to PLOS ONE. After careful consideration, we feel that it has merit but does not fully meet PLOS ONE’s publication criteria as it currently stands. Therefore, we invite you to submit a revised version of the manuscript that addresses the points raised during the review process.

We look forward to receiving your revised manuscript.

Kind regards,

Junzheng Yang

Academic Editor

PLOS One

“This work was partly supported by grant-in-aid funding from the Ministry of Education, Culture, Sports, Science, and Technology of Japan (#17H03806, #21H02117 and # 24K08715 to M. K.; #18K05393 to T. K.; #18K14377 to I. N.); the Mishima Kaiun Memorial Foundation to I. N.; the Science and Technology Research Promotion Program for Agriculture, Forestry, Fisheries, and Food Industry (#957613) to M. K.; we also thank the Faculty of Life and Environmental Sciences at Shimane University for the financial support for publishing this report.”

Reviewers' comments:

Reviewer's Responses to Questions

**Comments to the Author**

1. Is the manuscript technically sound, and do the data support the conclusions?

Reviewer #1: Yes

Reviewer #2: Yes

2. Has the statistical analysis been performed appropriately and rigorously?

Reviewer #1: No

Reviewer #2: Yes

3. Have the authors made all data underlying the findings in their manuscript fully available?

Reviewer #1: Yes

Reviewer #2: Yes

4. Is the manuscript presented in an intelligible fashion and written in standard English?

Reviewer #1: Yes

Reviewer #2: Yes

Reviewer #1: 1. The finding that the chromosomally integrated Pos5-GFP fusion did not localize correctly and was non-functional is critical but not fully explored. Was this due to the tag interfering with the mitochondrial targeting signal (MTS) or protein folding? This result should be discussed, as it serves as an important cautionary note for tagging strategies and confirms the necessity of the MTS.

2. The manuscript uses Student's t-test for all pairwise comparisons. When multiple strains or conditions are compared to a single control (e.g., *Δpos5+vector* in Fig 6), a one-way ANOVA with a post-hoc test (e.g., Tukey's) is more statistically appropriate. The authors should re-analyze their data using the correct statistical model or clearly justify the use of multiple t-tests.

3. The partial rescue by PHB is intriguing. The authors suggest it affects a reaction "upstream of CoQ biosynthesis." Since PHB is the direct substrate for Ppt1 (Coq2), does this imply that the Δpos5 strain has reduced PHB availability? Is the synthesis of PHB from tyrosine (or its mitochondrial import) potentially NADPH-dependent? This point warrants deeper discussion in the Results or Discussion section.

4. The use of "Orange" and "Green" bars in the graphs, with descriptions in the figure legends, is non-standard and can be confusing. A clearer approach would be to use direct labels on the graphs or within the figure panels themselves (e.g., "CoQ10 / cell" and "CoQ10 / vol"). Additionally, the y-axes for cell number and CoQ content should be more distinctly separated or represented in a dual-axis graph with clear labeling.

5. The authors correctly note that Pos5 is important but not essential, unlike core coq genes. However, they should more explicitly discuss whether the CoQ deficiency is a direct consequence of low NADPH for biosynthesis or an indirect effect of general mitochondrial dysfunction (e.g., impaired Fe-S cluster biogenesis, which is also Pos5-dependent). The data with the Δcyc1 mutant argue against a general respiratory defect, but other mitochondrial processes could be involved.

Reviewer #2: This is a well-conducted and clearly presented study that establishes a novel and important role for the mitochondrial NAD kinase Pos5 in coenzyme Q (CoQ) biosynthesis in both Schizosaccharomyces pombe and Saccharomyces cerevisiae. The authors provide compelling genetic and biochemical evidence showing that pos5 deletion mutants in both yeasts exhibit a significant (~80%) reduction in CoQ levels, accompanied by characteristic CoQ-deficient phenotypes. The demonstration of functional conservation through heterologous complementation, the requirement for mitochondrial localization of the NAD kinase activity, and the partial rescue by quinone precursors are particularly strong aspects of the work. The findings are novel and have potential implications for understanding human disorders related to CoQ deficiency.

Major Points:

(1)The authors conclude that Pos5/NADPH is required for an "earlier step" in CoQ biosynthesis, based on the partial rescue by PHB and vanillic acid (VA) and the lack of effect from coq6 overexpression. This is a reasonable hypothesis. However, to strengthen this conclusion, it would be informative to directly measure the levels of early intermediates (e.g., decaprenyl-PHB) in the Δpos5 mutant compared to wild-type and perhaps a coq6 mutant. This could more precisely pinpoint the bottleneck.

(2)Figure 3 shows a decrease in total mitochondrial NADP(H) in the S. pombe Δpos5 strain. The claim that "NADP+ showed the most pronounced reduction" is made in the text (Page 29), but the graphical presentation in Fig. 3A (stacked bars) makes it difficult for the reader to independently assess the relative changes in NADP+ vs. NADPH. Presenting these as separate bar graphs or including the numerical values in a supplementary table would enhance clarity and support the statement regarding species-specific differences with S. cerevisiae.

(3)The discussion proposes Atd1 as a potential NADPH-dependent link in the quinone precursor pathway (Page 30). This is an interesting speculation. Could the authors provide any preliminary data or genetic interaction (e.g., double mutant analysis, overexpression of atd1 in Δpos5) to test this hypothesis? If not, the text should more clearly frame this as a suggested model for future investigation.

(4) Several figures (e.g., Figs 1, 2, 4, 6-8) use a dual Y-axis format (cell number and CoQ content). While informative, the graphs are somewhat crowded. Ensuring high resolution and clear differentiation of data series in the final version is essential.

Figure 5 (localization) lacks a scale bar in the provided image. This must be added.

The labels "Orange bars" and "Green bars" in the figure legends refer to colors not visible in the grayscale PDF. Please use patterns (hatching, shading) or direct labeling (e.g., "left axis: CoQ10 per 10^9 cells; right axis: CoQ10 per 50 mL culture") to ensure accessibility.

(5) The manuscript states the use of Student's t-test. For all multi-group comparisons (e.g., Fig. 6, 7, 8), please confirm that the appropriate statistical test (e.g., ANOVA with post-hoc test) was applied where applicable, and specify which groups are being compared when significance is indicated.

Minor Points:

Abstract: The phrase "inability to grown on non-fermentable carbon sources" should be corrected to "inability to grow".

Page 9, Abstract & Page 19, Results: "CoQ10 level in ΔScpos5 were decreased..." should be "was decreased" or "levels were decreased".

Page 12: "Synthetic defied (SD) medium" should be "Synthetic defined (SD) medium".

Page 15: "...primers containing restriction sites from the S. pombe PR110 genome and the S. cerevisiae BY4741 genome as the template." The phrasing is slightly ambiguous. Consider: "...using the S. pombe PR110 genome and the S. cerevisiae BY4741 genome as templates, with primers containing restriction sites."

Page 19: "CoQ levels in the Δpos5 strain of our construct were decreased to 0.2-fold of thoes in the wild-type..." should be "those".

Page 30: "...sulfide accumulation is not enhanced CoQ deficiency." Should likely be "is not enhanced by CoQ deficiency."

References: The reference list appears comprehensive. Please ensure all in-text citations have a corresponding entry and vice-versa.

.

Reviewer #1: No

Reviewer #2: No

---

## [Author Response · Author response to Decision Letter 1]

26 Jan 2026

Thank you for evaluating our work and careful checking of our manuscript. We appreciate the positive responses from two reviewers. We marked the amended parts in red characters in the revised manuscript. We also extended the morphological analysis in Figure S2.

Reviewer #1

1) The finding that the chromosomally integrated Pos5-GFP fusion did not localize correctly and was non-functional is critical but not fully explored. Was this due to the tag interfering with the mitochondrial targeting signal (MTS) or protein folding? This result should be discussed, as it serves as an important cautionary note for tagging strategies and confirms the necessity of the MTS.

Answer: The improper localization and non-functionality of the chromosomally integrated Pos5-GFP protein is a matter of our interest too. Our immunoblotting analysis detected the Pos5-GFP fusion protein in a correct size, without seeing free GFP. The fusion protein appeared to be synthesized properly rather than being degraded. Therefore, we think the C-terminal GFP tagging caused steric hindrance. The GFP tag likely interferes the protein’s folding that would prevent correct localization and proper enzymatic function. We have added this point in the result (P18, L9 and 14).

2) The manuscript uses Student's t-test for all pairwise comparisons. When multiple strains or conditions are compared to a single control (e.g., *Δpos5+vector* in Fig 6), a one-way ANOVA with a post-hoc test (e.g., Tukey's) is more statistically appropriate. The authors should re-analyze their data using the correct statistical model or clearly justify the use of multiple t-tests.

Answer: Thank you for this suggestion. We have re-analyzed the data using a one-way ANOVA followed by Dunnett’s test for multiple comparisons against the control strain, instead of Student’s t-test. We confirmed the statistical significance using this appropriate analysis (Figures 1, 2, 4, 6, 7, and 8; Supplementary figures 3, 4, 5, 6 and 7; their legends; the method (P12, L4)).

3) The partial rescue by PHB is intriguing. The authors suggest it affects a reaction "upstream of CoQ biosynthesis." Since PHB is the direct substrate for Ppt1 (Coq2), does this imply that the Δpos5 strain has reduced PHB availability? Is the synthesis of PHB from tyrosine (or its mitochondrial import) potentially NADPH-dependent? This point warrants deeper discussion in the Results or Discussion section.

Answer: Knowledge of the PHB biosynthetic pathway in S. pombe is still limited. Atd1 is a candidate enzyme that converts 4-hydroxybenzaldehyde to PHB. We tested overexpression of the atd1 gene in the Δpos5 strain to see any effect on CoQ synthesis. The result showed slight increased CoQ levels in such a stain comparing with the one without the atd1 expression (Fig. S7), but the difference was not statistically significant. We can not clearly conclude the involvement of NADPH in the Atd1 reaction. We have included this data and discussion in the revised manuscript (P21, L21; P25, L6).

4) The use of "Orange" and "Green" bars in the graphs, with descriptions in the figure legends, is non-standard and can be confusing. A clearer approach would be to use direct labels on the graphs or within the figure panels themselves (e.g., "CoQ10 / cell" and "CoQ10 / vol"). Additionally, the y-axes for cell number and CoQ content should be more distinctly separated or represented in a dual-axis graph with clear labeling.

Answer: Thank you for this suggestion. We revised the figures to improve readability following your suggestion. We separated the data into two distinct panels; one for CoQ content per cell and one for CoQ content per culture volume. We replaced the color-based description with grayscale and made distinct patterns to ensure the figures are easily readable (Figures 1, 2, 4, 6, 7 and 8; Supplementary figures 3, 4, 5, 6 and 7).

5) The authors correctly note that Pos5 is important but not essential, unlike core coq genes. However, they should more explicitly discuss whether the CoQ deficiency is a direct consequence of low NADPH for biosynthesis or an indirect effect of general mitochondrial dysfunction (e.g., impaired Fe-S cluster biogenesis, which is also Pos5-dependent). The data with the Δcyc1 mutant argue against a general respiratory defect, but other mitochondrial processes could be involved.

Answer: While we cannot rule out a possibility that Pos5 deficiency indirectly affects CoQ levels via general mitochondrial dysfunction (e.g., impaired Fe-S cluster biogenesis), our data argue against this being the primary cause. Specifically, if the defect was solely due to the dysfunction of Fe-S cluster-dependent enzymes (such as Coq6, which requires ferredoxin and Arh1), supplementation of the upstream precursor PHB in the ∆pos5 strain should not rescue CoQ levels. The observation that PHB restores CoQ biosynthesis implies that the core biosynthetic machinery downstream of PHB is functional, and that the primary bottleneck lies in the supply of the precursor. We added this point in the revised manuscript (P24, L15)

Reviewer #2

1) The authors conclude that Pos5/NADPH is required for an "earlier step" in CoQ biosynthesis, based on the partial rescue by PHB and vanillic acid (VA) and the lack of effect from coq6 overexpression. This is a reasonable hypothesis. However, to strengthen this conclusion, it would be informative to directly measure the levels of early intermediates (e.g., decaprenyl-PHB) in the Δpos5 mutant compared to wild-type and perhaps a coq6 mutant. This could more precisely pinpoint the bottleneck.

Answer: This is an important point. We attempted to detect early intermediates, such as 3-decaprenyl-4-hydroxybenzoate (4-HP10), in the Δpos5 strain using LC-MS/MS. However, the levels of these intermediates were below the limit of detection under our current experimental conditions. Further optimization of extraction and detection methods is required to profile these trace intermediates, which remains to be a subject for future investigation.

2) Figure 3 shows a decrease in total mitochondrial NADP(H) in the S. pombe Δpos5 strain. The claim that "NADP+ showed the most pronounced reduction" is made in the text (Page 29), but the graphical presentation in Fig. 3A (stacked bars) makes it difficult for the reader to independently assess the relative changes in NADP+ vs. NADPH. Presenting these as separate bar graphs or including the numerical values in a supplementary table would enhance clarity and support the statement regarding species-specific differences with S. cerevisiae.

Answer: As suggested, we have modified Figure 3 to present the total NADP(H) levels and the specific redox states (NADP+ and NADPH) in separate graphs to allow for easier comparison (New Fig. 3).

3) The discussion proposes Atd1 as a potential NADPH-dependent link in the quinone precursor pathway (Page 30). This is an interesting speculation. Could the authors provide any preliminary data or genetic interaction (e.g., double mutant analysis, overexpression of atd1 in Δpos5) to test this hypothesis? If not, the text should more clearly frame this as a suggested model for future investigation.

Answer: We performed the suggested experiment. Overexpression of atd1 in the Δpos5 strain slightly increased CoQ levels, although it was not statistical significance (p > 0.05). We have included these results in Supplementary figure S7 and added this result in the result (P21, L21)

4) Several figures (e.g., Figs 1, 2, 4, 6-8) use a dual Y-axis format (cell number and CoQ content). While informative, the graphs are somewhat crowded. Ensuring high resolution and clear differentiation of data series in the final version is essential.

Figure 5 (localization) lacks a scale bar in the provided image. This must be added.

The labels "Orange bars" and "Green bars" in the figure legends refer to colors not visible in the grayscale PDF. Please use patterns (hatching, shading) or direct labeling (e.g., "left axis: CoQ10 per 10^9 cells; right axis: CoQ10 per 50 mL culture") to ensure accessibility.

Answer: Thank you for this suggestion. Please refer to our response to Reviewer #1, Point 4. We separated the graphs to resolve the crowding issue.

5) The manuscript states the use of Student's t-test. For all multi-group comparisons (e.g., Fig. 6, 7, 8), please confirm that the appropriate statistical test (e.g., ANOVA with post-hoc test) was applied where applicable, and specify which groups are being compared when significance is indicated.

Answer: Please refer to our response to Reviewer #1, Point 2. We re-analyzed the data using ANOVA followed by Dunnett’s test.

Minor Points:

Thank you for careful checking of our manuscript. We corrected according to your suggestion.

Abstract: The phrase "inability to grown on non-fermentable carbon sources" should be corrected to "inability to grow".

Answer: We corrected (P2, L13).

Page 9, Abstract & Page 19, Results: "CoQ10 level in ΔScpos5 were decreased..." should be "was decreased" or "levels were decreased".

Answer: We corrected (P2, L11; P20, L17).

Page 12: "Synthetic defied (SD) medium" should be "Synthetic defined (SD) medium".

Answer: We corrected (P5, L16).

Page 15: "...primers containing restriction sites from the S. pombe PR110 genome and the S. cerevisiae BY4741 genome as the template." The phrasing is slightly ambiguous. Consider: "...using the S. pombe PR110 genome and the S. cerevisiae BY4741 genome as templates, with primers containing restriction sites."

Answer: We corrected (P8, L15).

Page 19: "CoQ levels in the Δpos5 strain of our construct were decreased to 0.2-fold of thoes in the wild-type..." should be "those".

Answer: We corrected (P17, L10).

Page 30: "...sulfide accumulation is not enhanced CoQ deficiency." Should likely be "is not enhanced by CoQ deficiency."

Answer: We corrected (P24, L11).

References: The reference list appears comprehensive. Please ensure all in-text citations have a corresponding entry and vice-versa.

Answer: We checked all citations.

---

## [Decision Letter · Decision Letter 1]

10 Mar 2026

Dear Dr. Kawamukai,

Thank you for submitting your manuscript to PLOS ONE. After careful consideration, we feel that it has merit but does not fully meet PLOS ONE’s publication criteria as it currently stands. Therefore, we invite you to submit a revised version of the manuscript that addresses the points raised during the review process.

We look forward to receiving your revised manuscript.

Kind regards,

Junzheng Yang

Academic Editor

PLOS One

Journal Requirements:

Reviewers' comments:

Reviewer's Responses to Questions

**Comments to the Author**

Reviewer #1: All comments have been addressed

Reviewer #3: (No Response)

2. Is the manuscript technically sound, and do the data support the conclusions?

Reviewer #1: Yes

Reviewer #3: Yes

3. Has the statistical analysis been performed appropriately and rigorously?

Reviewer #1: Yes

Reviewer #3: Yes

4. Have the authors made all data underlying the findings in their manuscript fully available?

Reviewer #1: Yes

Reviewer #3: Yes

5. Is the manuscript presented in an intelligible fashion and written in standard English?

Reviewer #1: Yes

Reviewer #3: Yes

Reviewer #1: (No Response)

Reviewer #3: The manuscript is clearly written and well illustrated. The conclusions are supported by the results. The authors seem to have adressed most reviewers’ comments. I only have a few minor comments.

P3 L9 : the claim that « most prokaryotes synthesize CoQ endogenously » is incorrect, as CoQ biosynthesis is restricted to bacteria belonging to the phylum Pseudomonadota (doi : 10.1101/2025.09.17.676790). All other bacterial phylla, nor archea, do not produce CoQ.

P10 L18 : « OD600 = 1 point ». This seems strange. You can use « OD600 = 1 »

P10 L19 « Mitochondria were isolated according to a method described previously [27] with slight modification. ». The modifications should be indicated.

Fig 2A : red is used to display conserved residues between the two sequences. Therefore, the amino acids of two insertions which are found only in the S. cerevisiae sequence should be displayed in a different color than red.

Figure legends are long and repetitive (for example fig 1A-B, fig 2B-C, fig 6A-B, fig 7A-B). Many details could be placed in the methods section to shorten the figure legends.

P14 L3 : when first talking about UTR1, please explain the function of this protein.

P20 L14-16 : « Coq6 catalyzes C5-hydroxylation of the quinone precursor and requires reducing equivalents from NAD(P)H, through ferredoxin and ferredoxin reductase [35, 36]. » References 35 and 36 do not support this claim. The papers that show this are the following doi : 10.1016/j.chembiol.2011.07.008 and 10.1002/cbic.202300738

P20 L21 « However, CoQ levels of per cells and per volume » delete « of »

P21 L13 « clearly restored the CoQ a. » please correct to « clearly restored the CoQ level. »

.

Reviewer #1: No

Reviewer #3: No

---

## [Author Response · Author response to Decision Letter 2]

14 Mar 2026

The manuscript is clearly written and well illustrated. The conclusions are supported by the results. The authors seem to have addressed most reviewers’ comments. I only have a few minor comments.

Thank you for careful checking of our revised manuscript. Your comments help to improve the accuracy of the description. We answered the comments and explained the revised points.

P3 L9 : the claim that « most prokaryotes synthesize CoQ endogenously » is incorrect, as CoQ biosynthesis is restricted to bacteria belonging to the phylum Pseudomonadota (doi : 10.1101/2025.09.17.676790). All other bacterial phylla, nor archea, do not produce CoQ.

Ans. As you pointed out, we changed this part to ‘’belonging to the phylum Pseudomonadota’ and added the reference

P10 L18 : « OD600 = 1 point ». This seems strange. You can use « OD600 = 1 »

Ans. Yes, we changed so (P10 L17).

P10 L19 « Mitochondria were isolated according to a method described previously [27] with slight modification. ». The modifications should be indicated.

Ans. We added the description of the modified point in P10, L19.

Fig 2A : red is used to display conserved residues between the two sequences. Therefore, the amino acids of two insertions which are found only in the S. cerevisiae sequence should be displayed in a different color than red.

Ans. We changed according to your suggestion (New Figure2A).

Figure legends are long and repetitive (for example fig 1A-B, fig 2B-C, fig 6A-B, fig 7A-B). Many details could be placed in the methods section to shorten the figure legends.

Ans. We agreed. We delete some sentences and moved to M&M. (Fig 1A-B, Fig 2B-C, Fig 6A-B, Fig 7A-B).

P14 L3 : when first talking about UTR1, please explain the function of this protein.

Ans. We added the description (P14 L1).

P20 L14-16 : « Coq6 catalyzes C5-hydroxylation of the quinone precursor and requires reducing equivalents from NAD(P)H, through ferredoxin and ferredoxin reductase [35, 36]. » References 35 and 36 do not support this claim. The papers that show this are the following doi : 10.1016/j.chembiol.2011.07.008 and 10.1002/cbic.202300738

Ans. You are right. We added the appropriate references (P20 L4).

P20 L21 « However, CoQ levels of per cells and per volume » delete « of »

Ans. We amended this part and also the other parts (P20 L16 and else)

P21 L13 « clearly restored the CoQ a. » please correct to « clearly restored the CoQ level. »

Ans. We amended this part (P20 L23).

---

## [Decision Letter · Decision Letter 2]

17 Mar 2026

Mitochondrial NAD kinase Pos5 is required for CoQ biosynthesis in yeasts

PONE-D-25-62406R2

Dear Dr. Kawamukai,

We’re pleased to inform you that your manuscript has been judged scientifically suitable for publication and will be formally accepted for publication once it meets all outstanding technical requirements.

Kind regards,

Junzheng Yang

Academic Editor

PLOS One

Additional Editor Comments (optional):

Reviewers' comments:

Reviewer's Responses to Questions

**Comments to the Author**

Reviewer #3: All comments have been addressed

2. Is the manuscript technically sound, and do the data support the conclusions?

Reviewer #3: Yes

3. Has the statistical analysis been performed appropriately and rigorously?

Reviewer #3: Yes

4. Have the authors made all data underlying the findings in their manuscript fully available?

Reviewer #3: Yes

5. Is the manuscript presented in an intelligible fashion and written in standard English?

Reviewer #3: Yes

Reviewer #3: The authors adressed my comments. The manuscript has been improved and is now ready for publication.

.

Reviewer #3: No

---

## [Editor Report · Acceptance letter]

PONE-D-25-62406R2

PLOS One

Dear Dr. Kawamukai,

I'm pleased to inform you that your manuscript has been deemed suitable for publication in PLOS One. Congratulations! Your manuscript is now being handed over to our production team.

Kind regards,

on behalf of

Director Junzheng Yang

Academic Editor

PLOS One